# Effect of Polyphenolic Complements on Cognitive Function in the Elderly: A Systematic Review

**DOI:** 10.3390/antiox11081549

**Published:** 2022-08-10

**Authors:** María José Rodrigo-Gonzalo, Susana González-Manzano, Roberto Mendez-Sánchez, Celestino Santos-Buelga, Jose Ignacio Recio-Rodríguez

**Affiliations:** 1Grupo de Investigación de Polifenoles (GIP-USAL), Universidad de Salamanca, E-37007 Salamanca, Spain; 2Facultad de Enfermería y Fisioterapia, Universidad de Salamanca, E-37007 Salamanca, Spain; 3Grupo de Fisioterapia, Recuperación Funcional y Ejercicio Terapéutico del Instituto de Investigación Biomédica de Salamanca (IBSAL), E-37007 Salamanca, Spain; 4Unidad de Investigación en Atención Primaria de Salamanca (APISAL), Instituto de Investigación Biomédica de Salamanca (IBSAL), Red de Investigación en Cronicidad, Atención Primaria y Promoción de la Salud (RICAPPS), E-37007 Salamanca, Spain

**Keywords:** flavonoids, flavanols, stilbenes, cognitive function, cognitive impairment and neurophysicological function

## Abstract

Polyphenols have been shown to be effective against many chronic diseases. These compounds could have a beneficial effect at the cognitive level. The exact mechanism by which they provide positive effects at the cognitive level is not well known, but it is believed that they could counteract neuroinflammation. The objective of this study is to review nutritional interventions that include foods or supplements rich in flavanols, flavonols, or stilbenes to the usual diet on cognitive deterioration in people over 50 years of age. Clinical trials published in PubMed and Web of Science from 1 March 2010 to 1 March 2020 were explored, from which 14 studies were selected. All of them showed some improvement after the intervention. In interventions with flavanols and stilbenes, relevant improvements have been observed both in healthy patients and in patients with established cognitive impairment. Most studies agree that the greatest benefits are found with high doses and longer duration treatments. The changes were fundamentally assessed through cognitive tests, and in some of the studies, through magnetic resonance imaging (fMRI). The type of cognitive test used to assess the effect of the intervention was revealed to be critical. Several studies have also shown improvements in analytical parameters and blood pressure.

## 1. Introduction

The prevalence of neurodegenerative diseases and consequently mild cognitive impairment and dementia is increasing, especially in Western countries. There are currently an estimated 50 million people with dementia worldwide and that number is expected to reach 152 million by 2050 [1].

Thirty-five per cent of dementia cases are attributable to risk factors related to lifestyles that are potentially modifiable [2]. Nutrition influences brain structure, functional connectivity, and cognition, and may modulate the rate and extent of disease progression [3,4]. Various phytochemical compounds are being studied as potential agents for the prevention of chronic diseases, among them, the growing interest in polyphenols stands out [5].

Observational data suggest that moderate and regular intake of polyphenol-rich foods provides cognitive benefits. Evidence from animal studies, especially rodents, identifies a positive impact on cognitive outcomes after supplementing the diet with foods rich in polyphenols. The researchers primarily observed improvements in learning, working memory, spatial memory, and memory acquisition and retention [6,7]. However, there is a lack of evidence when it comes to humans. Some studies have shown a positive association between the intake of polyphenols and the improvement in cognitive performance [8,9,10,11], both in healthy adults [8] and in adults with mild cognitive impairment already established [9]. Simultaneously, cardiovascular and metabolic improvements are also observed [8,9]. However, not all results show a clear improvement [12,13]. The discrepancy in the results of some studies might be due to the diversity in terms of the phenolic profile of the food supplements, the dose of the polyphenolic compounds, and other aspects derived from the study itself, such as the size of the sample, the methodological design, the duration, the outcome variables, and the time established to determine the effect on cognitive performance, among others.

The exact mechanisms through which polyphenols can influence the cognitive process remain unknown. The consumption of compounds from the flavanol and stilbene families, present in a diversity of plant-based foods, such as tea, cocoa, grapes, and red wine [14,15,16], could improve the bioavailability of nitric oxide. This compound produces an increase in vasodilation and subsequently in cerebral perfusion. It is also believed that the improvement in cognitive function could be due to direct mechanisms, possibly counteracting neuroinflammation and oxidative stress, and to indirect mechanisms involving, among others, the modulation of the microbiota-gut-brain axis [17,18,19].

The objective of this systematic review is to know the effect of nutritional interventions that include foods or supplements rich in flavanols, flavonols, or stilbenes to the usual diet on cognitive deterioration in people over 50 years of age.

## 2. Materials and Methods

### 2.1. Eligibility Criteria

This study has followed the recommendations stated in the Preferred Reporting Items for Systematic Reviews and Meta-Analyses (PRISMA) guidelines. The systematic review protocol has been registered in PROSPERO (registration ID number CRD42022345081).

Only randomized controlled trials have been considered in this review. Nutritional interventions that included foods or supplements rich in flavanols, flavonols, or stilbenes, and in which a cognitive impairment variable was also measured, were selected. The intervention could be either a single dose or a chronic one and the target population was adults over 50 years of age. Trials whose intervention was not primarily aimed at measuring variables of cognitive impairment were excluded. The language of the articles could be English or Spanish.

### 2.2. Information Sources and Search Terms

A review of articles was carried out in two databases, PubMed and Web of Science. Based on the research hypothesis, i.e., foods or supplements rich in polyphenolic compounds incorporated into the usual diet could prevent cognitive impairment in older people, a search was carried out specifying those polyphenols with greater importance in that deterioration according to the literature (flavanols, flavonols, and stilbenes). The terms used were “Catechins OR Flavanols”, “Flavonols OR Quercetin OR Quercetine”, “Resveratrol OR Stilbenes” and “cognitive impairment”, “cognitive function”, “cognitive task”, “neuroinflammation”, “cognitive impairment”, “delayed memory”, “cognitive performance”, “neurophysicological function” OR “neurodegenerative disease”. Studies published between 1 March 2010 and 1 March 2020 were examined. Additional studies were included from the review of the references cited in the selected publications. The final search strategy used to select the articles can be seen in Table 1.

### 2.3. Data Extraction

The studies were first examined by title, then by abstract and, finally, by reading the full text (Figure 1), verifying that they met all the inclusion criteria. The review was carried out separately by three investigators (SG/JR/MR).

From the selected publications, two tables were elaborated in which the following quantitative and qualitative data were included: authors, year of publication, characteristics of the sample (sample size, age, and morbidities), description of the interventions (number of doses, time of follow-up, type and amount of supplements or food) and results of the cognitive variables. In addition, information was extracted on the objectives and main conclusions of each study (Table 2 and Table 3).

### 2.4. Quality Evaluation

The validity of the trials included in this work was independently assessed by three authors, using the Cochrane Collaboration tool to assess risk of bias [28]. The following domains were included: selection bias (sequence generation and allocation concealment), performance bias (blinding of participants and personnel), detection bias (blinding of outcome assessors), attrition bias (incomplete outcome data), reporting bias (selective reporting of outcomes), and other biases. After this assessment, each study was scored as having low, unclear, or high risk of bias. In case of disagreement among the three evaluators, a consensus was reached by discussion (Table 4).

## 3. Results

### 3.1. Characteristics of Included Studies

The characteristics of the studies included in this review (participants, intervention, follow-up, and main findings) are shown in Table 2 and Table 3.

The follow-up period varies in the different studies (Table 2). In one of the studies, follow-up is completed at one month [24], in others at two [8,9], three [10,11,20], four [21], or six months [13,22,25,26], and in one of them the last evaluation was carried out twelve months after the start of the intervention [12]. The number of evaluations carried out by the studies ranged from either two [8,9,10,11,13,20,21,22,23,25], three [26,27], or five evaluations [12,24].

The mean age of the participants was above 57 years in all the studies, except for three in which the datum is not reflected [9,24]. In one study, only women were included [11], and in two other studies, the number of subjects of each sex was not specified [9,10]. Participants were usually from healthy populations, except for five studies involving patients with mild cognitive impairment [9,12,13,21,22]. The number of participants in each study ranged from 10 [22] to 90 individuals [8] (Table 2).

The type of polyphenols used in the intervention to later evaluate their influence on cognitive deterioration varied in the different studies. In eight of them, the intervention was carried out with compounds from the flavanol family [8,9,12,20,21,22,23,24], and in six, resveratrol was used, belonging to the stilbene family [10,11,13,25,26,27]. However, no randomized clinical trial was exclusively performed with compounds belonging to the flavonol family, and their influence on cognitive impairment was found. Two of the works found were conducted using compounds from the family of flavonols and stilbenes together; 200 mg of resveratrol (stilbene) and 320 or 350 mg of quercetin (flavonol) daily [13,25]. In some studies, the intervention was carried out with pure polyphenolic compounds [8,9,10,11,13,23,25,27], but in others green tea extracts [12], cocoa [20], grapes [21,22], or even multi-ingredient supplements [26] were employed (Table 3).

In studies where the same type of polyphenols were assessed, different intervention amounts were used [8,9,23]. In most studies, a placebo was also included [10,11,12,13,21,24,25,26].

In addition, one of the studies performed a double intervention, i.e., different amounts of the polyphenolic compound and performance of physical exercise or not [20]. In studies conducted with compounds from the flavanol family, the amount of pure compound in patients in the intervention group ranged from 23 mg [23] to 993 mg per day [8]. For the studies conducted with resveratrol and quercetin in mixture with resveratrol, the amount of pure compound ranged from 75 mg [27] to 1000 mg per day [10]. In the studies whose intervention was carried out with food extracts, the amounts ranged between 2 g/day of green tea [12] and 72 g/day of freeze-dried grapes [21] (Table 3).

All studies used validated cognitive tests as a measurement instrument. Among them, the Minimental State Examination (MMSE), the Trail Making Test (TMT), the Verbal Fluency Test (VFT), the Montreal Cognitive Assessment (MOCA), the Rey Auditory Verbal Learning Test (RAVLT), the Wechsler Adult Intelligence Scale—III and IV (WAIS-III and WAIS IV), the Cognitive Failures Questionnaire (CFQ), the Stroop test, the Modified-Benton (ModBent), the Hopkins Verbal Learning Test—revised (HVLT-R), the FAS Word Fluency (COWA), the Alzheimer disease assessment scale cognitive (ADAS-Cog test), the Benton Visual Retention Test, the Boston Naming Test, the Rey Osterrieth complex figure test (ROCF), the Wisconsin Cart Sorting Test (WCST), the Cambridge Semantic Memory Battery, and the Eriksen flanker task. Some of the studies also used imaging tests, such as Magnetic Resonance Imaging (MRI) [13,20,21,25], Brain Computed Tomography [22], and cerebral blood flow analysis [11,23,27] to assess post-intervention changes (Table 3).

### 3.2. Effects of the Intervention with Flavanols

There are eight studies that were carried out with compounds from the flavanol family. In four of them, the population group was patients with mild cognitive impairment [9,12,21,22] and in the other four, the participants were healthy [8,20,23,24].

Several studies in which participants had cognitive impairment [9,12] concluded that there were no significant changes in Mini Mental State Examination (MMSE) scores after the intervention. This test evaluates temporal and spatial orientation, fixation, attention and calculation capacity, memory, nomination, repetition, compression, reading, writing, and drawing. However, in both cases, significant changes were observed, specifically after the intake of high (900 mg/day) and intermediate (520 mg/day) amounts of flavanols for eight weeks, when the Trail Making Test (TMT A and B) that evaluates attention, psychomotor speed, and cognitive flexibility, and the Verbal Fluency Test (VFT) that evaluates cognitive and verbal functions were used. The time to complete Trail Making Test A and Trail Making Test B was significantly shorter (*p* ≤ 0.05) in participants who consumed high (38.10 ± 10.94 s and 104.10 ± 28.73 s, respectively) and intermediate amounts of flavanols (40.20 ± 11.35 s and 115.97 ± 28.35 s) compared with subjects taking 45 mg/day of flavanols (52.60 ± 17.97 s and 139.23 ± 43.02 s). In the VFT, it was observed that the score was significantly better (*p* ≤ 0.05) in the subjects who consumed a high amount of flavanols in relation to those who consumed a low amount (27.50 ± 6.75 vs. 22.30 ± 8.09 words for 60 s) [9].

Krikorian et al. (2012) observed that after ingestion for 16 weeks of 100% Concord grape juice (containing 425 mg/L of anthocyanins and 888 mg/L of procyanidins), at a daily dose depending on the body weight of the subject (range 6.3–7.8 mL/kg), no significant differences in performance existed on the California Verbal Learning Test-II (CVLT). This test assesses verbal memory and learning ability. However, a trend towards better performance was observed in the placebo group (13.8 vs. 12.4 in the placebo and intervention groups, respectively, *p* = 0.09), but the intervention group made fewer errors (7.16 vs. 5.03 errors in the placebo and intervention groups, respectively, *p* = 0.04). In the same study, through Functional Magnetic Resonance Imaging (fMRI), greater activity was observed in certain regions of the hemisphere, associated with vascular benefit, in patients who had drank 100% Concord grape juice for 16 weeks compared with the control group [21]. Through the evaluation carried out with the Wechsler Adult Intelligence Scale (WAIS III), improvement in attention and working memory (*p* = 0.04) was shown after the intake of 72 g/day of freeze-dried grape powder for 6 months (total polyphenol content of 495 mg/100 g, catechin 23.9 mg/kg, epicatechin 23 mg/kg) [22].

It was also observed that subjects who took 72 g/day of freeze-dried grape powder for 6 months showed no decline in the metabolism of the right posterior cingulate cortex and left superior posterolateral temporal cortex. However, the group of participants who took a placebo did show a decline in metabolism (−4.04, *p* = 0.01, in the right posterior cortex and −3.06, *p* = 0.04, in the left superior posterolateral temporal cortex); these regions are affected in the early stages of Alzheimer’s disease [22]. These data were obtained using standardized volume of interest (sVOI) methods and statistical parametric mapping (SPM) software performed through the analysis of a brain Positron Emission Tomography (PET) scan.

Through the Geriatric Depression Scale (GDS), used to assess the level of depressive symptoms, it was found that the intake of 100% Concord grape juice for eight weeks, in which 425 mg/L anthocyanins and 888 mg/L flavanols were given, at a daily dose depending on the subject’s body weight (range 6.3–7.8 mL/kg), did not influence mood (5.7 vs. 6.9, *p* = 0.36) [21].

Different analytical parameters were analyzed in two studies [9,12]. In one of them, the analytical parameters were determined at the baseline evaluation and at two months from the start of supplement consumption [9], and in the other study, they were determined at the baseline evaluation and at three, six, nine, and twelve months from the start of the intervention [12]. Both observed improvements in some of the parameters analyzed. Lower insulin resistance and lower glucose levels in blood were determined after ingestion for eight weeks of a beverage containing 900 mg flavanols/day and 520 mg flavanols/day [9]. Ide et al. (2016) found a decrease in malondialdehyde-modified low-density lipoprotein, a marker of oxidative stress, after ingestion of 2 g/day of green tea (220.2 mg catechins) for twelve months [12]. However, in other analytical parameters, such as total cholesterol, c-LDL, c-HDL, or triglycerides, Ide et al. did not observe changes [12], as did Desideri et al. 2012 [9]. Lipid peroxidation (F2 isoprostanes) and blood pressure improved after ingestion for two months of a drink with 900 mg of flavanols/day and with 520 mg of flavanols/day. In both doses, the improvement was greater in systolic blood pressure [9].

All the studies in which the participants did not have previous cognitive pathologies [8,20,23,24] showed some benefit in the parameters analyzed after the intake of the flavanol supplement. In most cases [8,20,23], cognitive performance improved after the intervention. In the Trail Making Test (TMT A and B), the response time was significantly shorter after ingestion for 8 weeks of a daily drink with 993 mg of flavanols (TMT A −8.57 ± 0.38 s, and TMT B −16.50 ± 0.8 s, *p* < 0.0001) and with 520 mg (TMT A −6.67 ± 0.45 s, and TMT B −14.20 ± 0.49 s, *p* < 0.0001). This improvement was not seen with the daily 48 mg flavanol drink (TMT A −0.77 ± 1.57, *p* = 0.63 and TMT B −1.10 ± 0.68, *p* = 0.12) [8]. In the Verbal Fluency Test (VFT), the performance improved after the ingestion of the three drinks with the different amounts of flavanols that were evaluated in the study (993 mg, 520 mg, and 48 mg), but there was greater improvement with the drink with 993 mg of flavanols (7.70 words ± 1.09, *p* < 0.0001) than with 520 mg (3.57 ± 1.23, *p* = 0.007) and with 48 mg (1.33 ± 0.45, *p* = 0.01) [8]. Through the tasks of the Modified-Benton (ModBent), in which perception, visual memory, and visuoconstructive abilities are evaluated, an improvement in cognitive performance of 630 ms was observed in patients with a high intake (900 mg) of flavanols and 138 mg of epicatechin daily for three months. The reaction time (RT) of ModBent in patients with high flavanol content was 2 ms, while for those with a low intake (10 mg) of flavanols and <2 mg of epicatechin, the reaction time was 2.6 ms (*p* = 0.038). This improvement in ModBent coincided with an increase in cerebral blood volume (CBV) of the dentate gyrus [20]. This improvement was also seen in the study by Lamport et al. (2015) with increased regional cerebral perfusion at two hours after acute ingestion of the high flavanol beverage, in this case 494 mg [23]. However, no differences were detected in MMSE test scores after intake of different amounts of flavanols (993 mg, 520 mg, or 48 mg) for eight weeks and their baseline [8]. Pass et al. (2013), through the Cognitive Drug Research (CDR), a cognitive battery that measures reaction time, numerical and spatial working memory, word and image memory, and recognition and episodic secondary memory, did not observe differences at the different evaluation times (baseline—1 h–2.5 h–4 h–30 days) after acute intake of 500 mg of flavanols, 250 mg, or placebo. There were also no changes in mood using the Bond and Lader Visual Analogue Scales self-assessment after acute treatment, but greater calmness was observed after thirty days of ingestion of 500 mg of flavanols (t: −2.36, *p* < 0.05) [24]. Brickman et al. (2014) also evaluated physical exercise, without finding an influence of this variable (exercise group RT: 2.35 ms, control group RT: 2.28 ms, *p* = 0.815) [20].

Regarding other clinical parameters, an improvement in insulin resistance and a decrease in blood pressure, glucose, and lipid peroxidation were observed after daily intake of a drink with 993 mg and 520 mg of flavanols for eight weeks (8).

### 3.3. Effects of Intervention with Stilbenes

Of the six studies with stilbenes, four of them using resveratrol and two using flavonol (quercetin) with an admixture of resveratrol, one of them included participants with cognitive pathology [13] and five of them included participants considered healthy [10,11,25,26,27].

The study with participants with cognitive impairment [13] showed the following results: after a daily intake of 200 mg of resveratrol and 350 mg of quercetin for twenty-six weeks and through the Rey Auditory Verbal Learning Test (RAVLT), no effect was found in the evaluation. The RAVLT provides information on immediate or short-term memory, final acquisition level, delayed recall, and long-term memory. However, through Magnetic Resonance Imaging (NMR), they observed that after the intervention, the volume of the hippocampus was preserved and at rest the functional connectivity of the hippocampus improved, which did not occur in the placebo group of the study. Indeed, a 4.2% decrease in left hippocampal volume was shown in the placebo group [13].

All the five studies in which the participants were healthy subjects [10,11,25,26,27] showed improvements after taking resveratrol. After a daily intake for twenty-six weeks of 200 mg of resveratrol and 320 mg of quercetin, divided into four capsules per day, greater “delayed recall” and “recognition” of words (+1.8 words, *p* = 0.01, and +1.6 words, *p* = 0.02, respectively) was measured through the RAVLT. “Delayed recall” is understood as the number of words correctly recalled after a 30 min delay (maximum 15 words), and “recognition” as the number of words correctly recognized, minus false positives, from a subsequent list of 45 words read aloud that comprise the 15 correct words and another 30 that are not correct (maximum 15). Learning ability (the sum of words correctly recalled during the five immediate learning trials, maximum score 75 words) improved significantly in both groups, but with a greater difference in the intervention group than in the placebo group (+5.3 words, *p* = 0.02, and + 4.5 words, *p* = 0.06, respectively). However, “word retention” in the intervention group, after a daily intake of 200 mg of resveratrol and 320 mg of quercetin for twenty-six weeks, was significantly lower than in the baseline interview (−0.9, *p* = 0.04). “Word retention” is the number of correct words recalled after the fifth trial (maximum 15 words) subtracted from those correctly recalled after the 30 min delay (maximum 15 words) [25].

After ingestion of 1000 mg/day of resveratrol for 90 days, the Trail Making Test (TMT A) showed greater psychomotor speed (8.79 s and 10.59 s less to complete the test than after consuming placebo or after consuming 300 mg/day of resveratrol, respectively, *p* = 0.02) [10]. Multi-ingredient supplementation (omega 3 polyunsaturated fatty acids, vitamin D, 150 mg of resveratrol, and whey protein) for six months showed a reduction in the response time to complete tasks, and better performance in the Stroop test at six months [26]. In addition, after acute intake of 75 mg, 150 mg, or 300 mg of resveratrol, the cerebral vasodilator response (CVR) in the Middle Cerebral Arteries (MCA) increased, measured at 45–60 min and 90–120 min post treatment. The improvement compared with the placebo group was 13.8 ± 3.5% in the intake of 75 mg of resveratrol (*p* = 0.001), 8.9 ± 3.5% in the intake of 150 mg of resveratrol (*p* = 0.02), and 13.7 ± 3.3% in the 300 mg of resveratrol (*p* < 0.001). In the Posterior Cerebral Arteries (PCA), only the 75mg dose of resveratrol was effective, 13.2 ± 4.5% (*p* = 0.02) [27]. Evans et al. (2017) reported that a 17% increase in CVR correlated with significant improvements in cognitive performance, significant improvements in the immediate RAVLT (+4.5 ± 1.3 words, *p* = 0.02), in categorical fluency (+3.2 ± 1.7 words, *p* = 0.02), and in the camel and cactus test (+2.9 ± 1.5 correct answers, *p* = 0.01). These improvements correlated with increased CVR (11). Witte et al. (2014) also demonstrated an increase in functional connectivity of the hippocampus in the frontal, parietal, and occipital areas after daily intake of 200 mg resveratrol and 320 mg quercetin for twenty-six weeks compared with the placebo group (*p* < 0.05) [25].

On the other hand, no significant changes in performance on tests of visual attention, working memory, verbal fluency, and semantic memory were observed between the 1000 mg resveratrol, 300 mg resveratrol, or placebo intervention over 90 days [10]. Kobe et al. (2017) showed that glycosylated hemoglobin (HbA1c) was significantly reduced after ingestion of 200 mg of resveratrol and 350 mg of quercetin for twenty-six weeks [13].

## 4. Discussion

### 4.1. Main Findings

In general, the regular intake of foods rich in polyphenols is related to cognitive, cardiovascular, and metabolic benefits. The present work tries to ascertain if interventions with foods or food supplements rich in flavonols, flavanols, and stilbenes improve cognitive capacity in people over 50 years of age. All the studies analyzed showed some improvement in health indicators after supplementation with flavanols or stilbenes. However, not all of them concluded a greater cognitive capacity [12,24]. No relevant articles related to the influence of flavonols on cognitive deterioration were found in the specific search run for them, although some of the studies analyzed did contain them in combination with other polyphenols.

Cognitive performance improved with supplementation of flavanols, stilbenes, and a mixture of stilbenes plus flavonols. It was observed that the “attention” measured by different tests (TMT A and B and WAIS III) improved both with flavanol [8,9,22] and with stilbene supplements [10]. This benefit was found both in patients with and without cognitive pathology. The “memory” was evaluated in different ways: after supplementation with flavonoids and/or stilbenes, visual memory, evaluated with the ModBent Test, improved in patients without cognitive pathology [20], and working memory, evaluated with the WAIS III, also significantly improved in patients with cognitive pathology [22]. However, working memory, assessed with the CDR test [24], and verbal memory, assessed with the CVLT test [21], showed no change after flavonoid supplementation, respectively. The difference in results depending on the type of memory analyzed, in addition to the fact that the polyphenolic supplement affected each of the memory types differently, might be due to the type of polyphenols used in each study; in the studies where improvements were found, flavanols were used together with other compounds belonging to other families of flavonoids [21,22], while in the studies in which no such benefit was found, the supplement only consisted of flavanols [24]. On the other hand, attention and immediate memory are parameters also evaluated in the MMSE [8,9,12], but in this case, no significant difference was observed in the total score of the test after the flavanol or stilbene supplements. Perhaps if the test scores were broken down into the different parameters evaluated, some differences could have been found. Psychomotor speed, assessed with TMT A and B, improved with both flavanol and stilbene supplementation, and both in patients with and without cognitive pathology. Delayed recall and word recognition, after supplementation with stilbenes and flavonoids, improved in patients without cognitive impairment (supplementation of 200 mg resveratrol and 320 mg quercetin) [25], but no differences were observed in patients with cognitive pathology (after the supplementation of 200 mg of resveratrol and 350 mg of quercetin) [13]. These differences could be due to the fact that similar interventions have different effects, depending on whether one has a cognitive pathology or not.

The choice of the cognitive test to assess post-intervention changes is key. In all studies using the Trail Making Test (TMT A and B) [8,9,10] and the Verbal Fluency Test (VFT) [8,9], cognitive improvements at the higher doses used in the intervention were found. However, in all the studies that used the MMSE Test [8,9,12,22], no significant post-intervention changes were observed. Through the Modified-Benton (ModBent) tasks, the Wechsler Adult Intelligence Scale III (WAIS-III), and the Stroop tests, cognitive improvements were also observed after the intervention [[20],[22],[26] respectively]. With the Rey Auditory Verbal Learning Test, in the study in which subjects with cognitive pathology participated, no changes were observed after the daily intake of 200 mg of resveratrol and 350 mg of quercetin for twenty-six weeks [13]. However, significant changes, with greater retention of words, were determined in patients without cognitive pathology who took 200 mg of resveratrol and 320 mg of quercetin daily for twenty-six weeks [25]. Therefore, each cognitive test evaluates specific cognitive functions, and depending on which one is employed, the results will be different, since each type of polyphenol could have a specific impact on each of the functions. The measure of dispersion of the results of a test is also important; indeed, the precision of the test is a very important factor to take into account when selecting a cognitive test. Through imaging tests, all studies showed a greater cerebral vasodilator response and greater connectivity, which is related to better cognitive capacity [13,20,21,22,23,27].

If the evolution of the analytical parameters is considered, some metabolic improvement was found in all the studies measuring them [8,9,13]. Reductions in blood glucose, insulin resistance, and lipid peroxidation were improvements observed in patients who added a flavanol supplement to their diet, both with and without cognitive pathology [8,9]. However, the level of low-density lipoprotein (LDL) decreased after the intake of 2 g/day of green tea (220.2 mg of catechins) for twelve months [12], but not after the intake of 900 mg, 520 mg, or 45 mg of flavanols for eight weeks [9]. Perhaps this discrepancy between the two studies in terms of LDL cholesterol may be influenced by the type of polyphenolic compound used in the intervention and by the time of duration of the intervention. Flavanols derived from epicatechin (monomers, dimers, and trimers) are found in cocoa, while flavanols derived from gallocatechins (e.g., epigallocatechin-3-gallate (EGCG), epigallocatechin, gallocatechin, and gallocatechin gallate) predominate in tea. This latter group possess one hydroxyl more than epicatechin and can also be esterified with gallic acid, which could influence their activity. It must also be noted that both the amount of polyphenol consumed and the duration of the intervention were different in the two studies (two months vs. twelve months). Therefore, the time at which the analysis is performed can also influence the results. On the other hand, a decrease in glycosylated hemoglobin was observed after daily intake of 200 mg of resveratrol and 350 mg of quercetin for twenty-six weeks in patients with cognitive pathology [13].

Blood pressure was analyzed in two studies. In both, their value decreased after the intervention with the highest doses of the intervention with flavanols [8,9]. Both studies observed greater improvement in SBP with very similar flavanol values.

### 4.2. Strengths and Limitations

In general, the studies analyzed in this review do not provide detailed information on the participants, level of education, and socioeconomic status, which could also have an influence on the results. In addition, this hinders reproducibility and proper assessment of the rigor of study design and execution.

Another important limitation of the studies is their short follow-up period. Only in one of them, the last evaluation was carried out twelve months after the start of the intervention [12], and in the rest of the studies, shorter follow-up periods were considered, which limits information on how long the improvements are maintained (i.e., long-term effects).

Furthermore, some studies have a small sample size, resulting in low statistical power and making it difficult to demonstrate the efficacy of the interventions performed.

Finally, in this review, only articles published in English or Spanish were considered and no unpublished literature was sought, nor publications not collected in PubMed and Web of Science, so potentially interesting studies could have been omitted.

### 4.3. Comparison with Other Reviews

There are several reviews on polyphenols and chronic diseases, however, not so many related to the influence of cognitive decline and polyphenols. Most beneficial effects are seen with the higher amounts of polyphenol used in the intervention groups [29,30]. Furthermore, the intake of certain classes of polyphenols, rather than total polyphenols, seems to be a key factor in reducing the incidence of various chronic diseases. In particular, distinct studies reported positive results after the intake of flavonols, flavones, flavanones, isoflavones, anthocyanidins, and/or proanthocyanidins, especially in diseases such as diabetes, heart disease, and kidney failure, and in respiratory and immune function, as well as in the prevention/delay of cognitive decline, brittleness, or bone fractures [29,31].

Various investigations point to the Mediterranean diet as a protective factor against cognitive deterioration, associated with a high consumption of vegetables, fruits, nuts, and olive oil, which are rich sources of polyphenolic compounds [32,33,34,35]. Other reviews also conclude that flavonoids, present in fruits, cocoa, wine, tea, and beans, may be related to better cognitive evolution [36]. Another piece of information in common is regarding the MMSE Test as a tool with limitations to be used in research [32]. As concluded by Lamport et al. (2021), a positive association between the consumption of polyphenols and cognitive capacity can be observed, but not definitively confirmed, owing to the methodological heterogeneity in the clinical trials carried out [37].

## 5. Conclusions

Dietary supplementation with flavanols or stilbenes has been related with improved attention, psychomotor speed, delayed recall, and word recognition. The synergistic action of different types of flavonoids has proven to be meaningful to improve visual memory in patients without cognitive pathology and working memory in patients with cognitive pathology. No significant changes have been observed in working and verbal memory after the interventions.

It can be concluded that, in all the studies analyzed, some improvement in health after the intervention with flavanols, stilbenes, and flavonoids has been found. In general, more positive results are observed in studies in which a greater amount of polyphenols is consumed. The duration of the intervention and the period in which the evaluation is carried out also have an influence; the longer the intervention period, the more positive results are generally observed.

Despite the findings made, further research is necessary to be able to define more specific recommendations on the intake of these components, in terms of quantity and type of polyphenol, as well as to elucidate the mechanism of action behind their activity in humans.

## Figures and Tables

**Figure 1 antioxidants-11-01549-f001:**
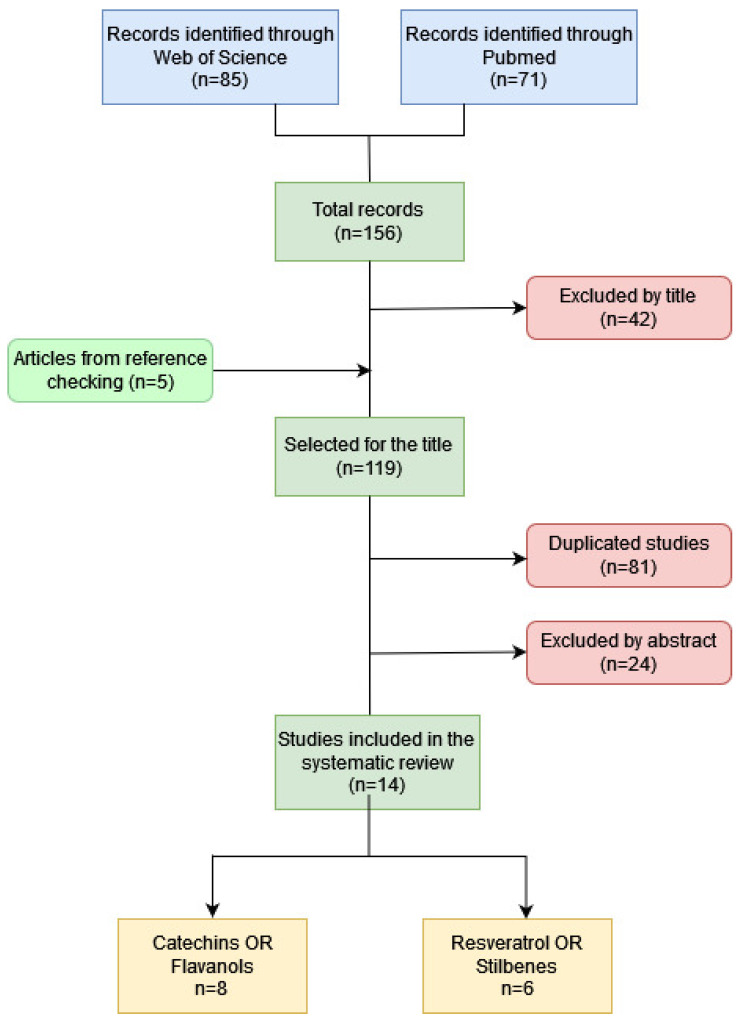
Study flow diagram: search strategy.

**Table 1 antioxidants-11-01549-t001:** Search strategy.

Database	Combination	Results	Selection by Title	Duplicates	Selection by Abstract
Web of Science	(Catechins OR Flavanols) AND (“cognitive impairment” OR “cognitive function” OR “cognitive task” OR “neurodegenerative disease” OR neuroinflammation OR “neurophysicological function”)	11	11	0	5
	(Catechins OR Flavanols) AND (“cognitive impairment” OR “delayed memory” OR “cognitive performance”)	16	16	10	1
PubMed	(Catechins OR Flavanols) AND (“cognitive impairment” OR “cognitive function” OR “cognitive task” OR neuroinflammation)	11	10	10	-
	(Catechins OR Flavanols) AND “neurodegenerative disease”	0	0	0	0
	(Catechins OR Flavanols) AND “neurophysicological function”	0	0	0	0
	(Catechins OR Flavanols) AND (“cognitive impairment” OR “delayed memory” OR “cognitive performance”)	9	9	9	-
Web of Science	(Flavonols OR Quercetin OR Quercetine) AND (“cognitive impairment” OR “cognitive function” OR “cognitive task” OR “neurodegenerative disease” OR neuroinflammation OR “neurophysicological function”)	9	7	4	1
	(Flavonols OR Quercetin OR Quercetine) AND (“cognitive impairment” OR “delayed memory” OR “cognitive performance”)	12	10	9	1
PubMed	(Flavonols OR Quercetin) AND (“cognitive impairment” OR “cognitive function” OR “cognitive task” OR neuroinflammation)	5	5	5	-
	(Flavonols OR Quercetin) AND (“neurodegenerative disease” OR “neurophysicological function”)	0	0	0	0
	(Flavonols OR Quercetin OR Quercetine) AND (“cognitive impairment” OR “delayed memory” OR “cognitive performance”)	6	6	6	-
Web of Science	(Resveratrol OR Stilbenes) AND (“cognitive impairment” OR “cognitive function” OR “cognitive task” OR “neurodegenerative disease” OR neuroinflammation OR “neurophysicological function”)	18	10	2	4
	(Resveratrol OR Stilbenes) AND (“cognitive impairment” OR “delayed memory” OR “cognitive performance”)	19	11	8	1
PubMed	(Resveratrol OR Stilbenes) AND (“cognitive impairment” OR “cognitive function” OR “cognitive task” OR neuroinflammation)	24	10	9	0
	(Resveratrol OR Stilbenes) AND (“neurophysicological function”)	0	0	0	0
	(Resveratrol OR Stilbenes) AND (“neurodegenerative disease”)	0	0	0	0
	(Resveratrol OR Stilbenes) AND (“cognitive impairment” OR “delayed memory” OR “cognitive performance”)	16	9	9	-
	Additional bibliography obtained from the articles		5	0	1

**Table 2 antioxidants-11-01549-t002:** Characteristics of included studies.

Study	Number of Participants	Pathologies of the Participants	Selection Criteria	Follow-Up (Months)
**Catechins/Flavanols**				
Ide et al. (2016) [12]	33 (27 at the end) I: 17 (14) C: 16 (13)	Yes -17 Alzheimer’s -15 Insanity -1 Lewy body dementia	-Over 50 years. -Do not consume antioxidant stress supplements (vit E, C, A, and B-carotene). -MMSE (Japanese version) < 28 -No allergy to tea. -Without severe cardiac, respiratory, hepatic, or renal dysfunction or anemia.	12
	29 women 4 men Mean age: 84.8 y			
Mastroiacovo et al. (2015) [8]	90 (86 at the end) -High intake: 30 (29) -Average intake: 30 (29) -Low intake: 30 (28)	No	-No smoking. -Do not consume supplements (vit C and E) and/or medications with antioxidants (statins and glitazones), or drugs that interfere with consciousness (benzodiazepines and antidepressants), or chocolate or cocoa. -MMSE ≥ 27 and geriatric depression scale < 11. -BMI ≤ 30 and no weight change (±10%) in the last 6 months. -No medical conditions (no cardiovascular disease, cerebrovascular event, neurological or thyroid disorders, or inflammatory diseases).	2
	37 women 53 men Median age: 69.6 y			
Brickman et al. (2014) [20]	37 -High intake + exercise: 8 -High intake + no exercise: 11 -Low intake + exercise: 9 -Low intake + no exercise: 9	No	-Do not consume dietary or herbal supplements. -No treatment with psychotropic drugs or current psychiatric disorder. -No diabetes Dx. -People who do not exercise regularly or do not exceed the American Heart Association standards for average physical fitness. -No contraindication to aerobic exercise. -Not lactose intolerant.	3
	28 women 9 men Median age: 57.8 y			
Desideri et al. (2012) [9]	90 -High intake: 30 -Intermediate: 30 -Low: 30	Yes -Mild Cognitive Impairment	-Dx mild cognitive impairment according to Petersen criteria. -Do not smoke or take statins. -No Dx obesity (BMI > 30).	2
	No differentiated data for women and men			
Krikorian et al. (2012) [21]	21 I: 10 C:11	Yes -Mild Cognitive Impairment	-Dx mild cognitive impairment. -No Dx or suspected dementia, diabetes, kidney disease, liver disease, serious psychiatric conditions, and substance abuse.	4
	10 women 11 men Average age men: 76.9 y Median age women: 76 y			
Lee et al. (2017) [22]	10 I: 5 C:5	Yes -Mild Cognitive Impairment	-Cognitive deficit and/or personality change present for at least 6 months but without a diagnosis of Alzheimer’s or other cause of dementia. Not have cognitive dysfunction that decreases the ability to perform daily activities. -No history of thyroid disease. -No auditory or motor deficit that prevents performing the tests. -No treatment for Alzheimer’s or dementia-related medications. -CT/MRI if stroke, tumor, hemorrhage, ictal activity, or hydrocephalus is suspected.	6
	5 women 5 men Average age 72.2 y			
Lamport et al. (2015) [23]	18 High intake: 9/9 Low intake: 9/9	No	-No chronic drug use. No high alcohol consumption (15 units/week) or illegal substances. -No neurological symptoms, MMSE ≥ 26. -No current or recent disease (obesity, diabetes, cardiovascular disease, hypertension, stroke, gallbladder or gastrointestinal problems). -No language or hearing impairments. -No allergy or sensitivity to chocolate, dairy, nuts, or gluten.	1 day Crossover study Washout: 1 week
	10 women 8 men Average age: 61 y			
Pase et al. (2013) [24]	71 -High intake: 24 -Average intake: 25 -Placebo: 22	No	-No smoking. -Do not consume vitamin supplements, herbal extracts or illicit drugs. -Not being pregnant or breastfeeding. -No heart disease, hypertension, anxiety, depression, psychiatric disorders or epilepsy, or any other health disorder that affects food metabolism, such as kidney, liver, or gastrointestinal disease.	1
	47 women 24 men			
**Resveratrol/Stilbenes**				
Witte et al. (2014) [25]	46 I: 23 -C:23	Yes -Overweight	-Not in treatment with antidepressants. -Do not consume a daily consumption of 50 g of alcohol, nor 10 cigarettes, nor 6 cups of coffee per day or more. -MMSE ≥ 26 points. -No serious medical, neurological, and psychiatric illnesses without treatment. -No Dx diabetes mellitus type 2. -Speak native German.	6 +1/2
	18 women 28 men Mean age group I: 64.8 y Mean age group C: 63.7 y			
Anton et al. (2018) [10]	32 -I (1000 mg): 10 -I (300 mg): 12 -C: 10	Yes -Overweight or obesity	-No smoking. -Do not consume anabolic drugs or anticholinesterase inhibitors or anticoagulants (the use of aspirin is allowed) or statins. Not on antidepressant treatment. -Not on active treatment for cancer, stroke (<6 months), peripheral vascular disease, coronary artery disease (myocardial infarction < 6 months), stage III or IV congestive heart failure, valvular heart disease, severe anemia, liver or kidney disease, diabetes, severe arthritis, limb fracture ≤ 6 months, or limb amputation. -Do not consume excessive alcohol (>2 drinks/day) or >1 glass of red wine or purple grape juice/week. -Not consuming or having consumed a dietary supplement containing resveratrol, quercetin, grape seed extract, ginko biloba, or *P. cuspidatum* in the previous 90 days. -No Dx dementia disease, severe psychiatric illness, or Parkinson’s disease. -No deterioration in vision and hearing. -Sedentary (less than 120 min of physical activity of moderate intensity per week). -Self-reported ability to walk one mile. -CES-D ≤ 20. -MMSE > 24 points. -BMI: 25–34.9 kg/m^2^. -Not a BP > 180/100 mmHg or a resting heart rate > 120 beats per minute. -Not having participated in another clinical trial or intake of an investigational product ≤ 30 days before the exam. -No contraindications for MRI.	3
	Average age: 73.34 y			
Moran et al. (2018) [26]	51 (37 at the end) I: 26 (20) C:23 (17)	No	-65 years or older. -Normal cognitive function (MMSE > 24). -Defined as “healthy” (free from disease). -Independent, mobile, and able to complete the test.	6
	19 women 18 men Average age: 75.14 y			
Wong et al. (2016) [27]	36 I (300 mg): 36/36 I (150 mg): 36/36 I (75 mg): 36/36 C: 36/36	Yes -Diabetes mellitus type 2	-Do not smoke or use nicotine therapy. -Not being on insulin or warfarin treatment. -No change in pre-existing medication or supplements during the intervention. -Not a modified MMSE (Australian version) <78/100. -No BP > 160/100 mmHg, no BMI > 40 mg/m^2^. -With the ability to obtain satisfactory images of the MCA bilaterally by TCD. -No serious head injuries. -No Dx of dementia, severe depression, stroke or neurological conditions; nor kidney or liver disease.	1 dayCross study. Wash: 1 week
	10 postmenopausal women 26 men Average age: 68.47 y			
Evans et al. (2017) [11]	80 (72 at the end) postmenopausal women I: 37 Q:35	No	-No smoking. -Not on insulin, warfarin, or hormone replacement therapy in the last six months. -No suspicion of dementia, modified MMSE (Australian version) ≥78 points. -No history of breast or cervical cancer. -No Dx of cardiovascular, renal, hepatic disease, depression or disorders.	3 + 1/2
	Median age: 61.5 y			
Kobe et al. (2017) [13]	40 I: 18 c:22	Yes -Mild Cognitive Impairment (diagnosed according to May criteria 12 months before)	-No MMSE < 24 at initial visit. -No BMI < 18 kg/m^2^ or >35 kg/m^2^. -Be right-handed. -Speak German fluently. -No serious untreated medical, neurological, or psychiatric illnesses and brain pathologies identified on MRI.	6 + 1/2
	21 women 19 men Mean age group I: 65 y Average age group C: 69 y			

The number in parentheses in the participant column corresponds to the number of participants at the end of the study. **Abbreviations:** BMI: Body Mass Index; BP: Blood Pressure; C: Control Group; CES-D: The Center for Epidemiological Studies Depression Scale; CT: Tomography Scan; Dx: Diagnostic; I: Intervention Group; MMSE: Mini-Mental Status Examination; MRI: Magnetic Resonance Imaging; TCD: Transcranial Doppler; Tto: Treatment.

**Table 3 antioxidants-11-01549-t003:** Summary of included studies.

Study	Intervention	Cognitive Variable Measurement Tool and Measurement Frequency (Months)	Results
**Catechins/Flavanols**			
Ide et al. (2016) [12]	-I: 2 g/day green tea (220.2 mg catechins) -C: placebo	-MMSE (Japanese version) -NPQ-I -Laboratory tests (MDA-LDL, TC, LDL, HDL, TG, FPG, HbA1c)	-The levels of MDA-LDL, a marker of oxidative stress, were significantly lower in the green tea group. -No significant changes in the rest of the analytical parameters. -No significant changes in MMSE-J or NPI-Q scores.
		(0–3-6–9–12 months)	
Mastroiacovo et al. (2015) [8]	-I: HF drink (993 mg/day flavanol) -I: IF drink (520 mg/day flavanol) -I: LF drink (48 mg/day flavanol)	-MMSE -TMT A and B -VFT -BP -Laboratory tests (glucose, insulin, TC, LDL, HDL, TG, HOMA-IR index, 8-Iso-PGF)	-Lower response time after the HF and IF drinks in the TMT tests. -Improvement in all treatments in the VFT test, but greater improvement after the HF drink. -Lower SBP and improvement in insulin resistance and fluid peroxidation after drinking HF and IF. -No change in MMSE score or DBP.
		(0–2 months)	
Brickman et al. (2014) [20]	-I: High intake (900 mg cocoa flavanols and 138 mg epicatechin daily) + aerobic exercise (1 h/day 4 days/week) -I: High intake + no aerobic exercise -I: Low intake (10 mg cocoa flavanols and 12 mg epicatechin daily) + aerobic exercise (1 h/day 4 days/week) -I: Low intake (10 mg cocoa flavanols and 12 mg epicatechin daily) + no aerobic exercise	-ModBent test -fMRI	-An intervention with high flavanol content has a significant effect on ModBent performance (independent of exercise). Improvement of 630 ms with respect to those of the intervention with low content. -Changes in the ModBent and changes in the CBV images of the dentate gyrus are correlated. -Exercise has no significant effect on ModBent.
		(0–3 months)	
Desideri et al. (2012) [9]	-I: High intake (990 mg/day flavanols) -I: Intermediate intake (520 mg/day flavanols) -I: Low intake (45 mg/day flavanols)	-MMSE -TMT A and B -VFT -BP -Laboratory tests (glucose, insulin, TC, LDL, HDL, TG, HOMA-IR index, 8-Iso-PGF)	-Lower response time after high intake drinks (−14.3 s and −29.2 s) and intermediate (−8 s and −22.8 s) in the TMT A and B tests, respectively. -Greater improvement in VFT in subjects with high drink intake (+8 words in 60 s) and medium (+5.1 words), and, to a lesser extent, in low intake (+1.2 words). -Lower insulin resistance in high (−1.6 mU/L) and medium (−0.9 mU/L) intake. -Lower glucose levels in high (−0.6 mmol/L) and medium (−0.5 mmol/L) intake. -BP: High intake (SBP −10 mmHg and DBP −4.8 mmHg), medium (SBP −8.2 mmHg and DBP −3.4 mmHg) and, to a lesser extent, low intake (SBP −1.4 mmHg and DBP −0.9 mmHg). -Total plasma levels of 8-iso-PGF2 decrease in high (−99.8 pg/L) and intermediate (−65.8 pg/L) intake. -No change in MMSE score. -No changes in insulin, total cholesterol, LDL, HDL, and triglycerides.
		(0–2 months)	
Krikorian et al. (2012) [21]	-I: Grape juice (425 mgl/L anthocyanins and 888 mg/L procyanidins) -C: Placebo * Daily dose by weight (6.3–7.8 mL/kg)	-GDS -RAVLT -CVLT -BP -Laboratory tests (glucose, insulin) -Anthropometric parameters (weight, height) -fMRI	-CVLT without changes, but with a tendency to better performance in the placebo group (13.8 vs. 12.4) but fewer errors (7.16 vs. 5.03) and better ability to distinguish the elements learned in the intervention group. -Greater activation in the regions of the hemisphere (fMRI images) of the patients in the group consuming grape juice, in turn being associated with a vascular benefit. -No changes in analytical or anthropometric parameters. -No changes in the GDS.
		(0–4 months)	
Lee et al. (2017) [22]	-I: freeze-dried grape powder made from fresh California grapes -C: placebo * 72 g/day, equivalent to 3 standard servings of fresh grapes per day.	-ADAS-Cog -MMSE -HVLT -Benton visual retention test -ROFT Test -Boston Appointments Test -Fluency of FAC letters -Fluency of categories (naming animals) -Stroop test -TMT A and B -WCST -Symbol and digit speed (WAIS-III tasks) -MFQ -HDRS -fMRI	-Improvements in attention/working memory according to the WAIS-III tasks in the intervention group. -Decline in the posterior cingulate and superior posterolateral temporal cortical regions in the placebo group while the intervention group did not observe said decline. -No significant changes in the rest of the tests.
		(0–6 months)	
Lamport et al. (2015) [23]	-High intake: 494 mg/day flavanols -Low intake: 23 mg/day flavanols	-CBF -fMRI	-Better regional cerebral perfusion in the group that consumed the drink with high flavanol intake (in the anterior cingulate cortex and in the parietal lobe regions).
		(0–2 h)	
Pase et al. (2013) [24]	-I: High intake (500 mg/day cocoa flavanols) -I: Average intake (250 mg/day cocoa flavanols) -C: Placebo	-CDR -Bond-Lader Visual Analogue Scale	-Greater calm and self-rated satisfaction after receiving 30 days of treatment with high dose of flavanols. -Mood did not change with acute treatment. -Cognitive performance did not vary at any of the measurement times.
		(0–1 h-2.5 h-4 h-30 days)	
**Resveratrol/Stilbenes**			
Witte et al. (2014) [25]	-I: 200 mg resveratrol and 320 mg quercetin daily -C: Placebo	-AVLT -PANAS -Freiburger Physical Activity Questionnaire -BP -Laboratory tests (glucose, insulin, TC, LDL, HDL, TG, HbA1c, leptin, BDNF, IGF-1, TNF-, interleukin 6, hs-CRP) -Anthropometric parameters (biomedical impedance, weight, height) -Vascular markers (CMIT)-fMRI	-Better delayed recall and recognition in the subjects of the intervention group. Learning ability significantly improved in both groups, but with a greater difference in the group that ingested resveratrol and quercetin. After the intervention, retention in the subjects was lower. * retention: number of correct words recalled after the fifth trial (maximum 15 words) subtracted from those recalled correctly after the 30 min delay (maximum 15 words). * delayed recall: number of words correctly recalled after the 30 min delay (maximum 15 words). * recognition: number of words correctly recognized minus false positives from a subsequent list of 45 words read aloud that included 15 correct words and 30 new ones (maximum 15). * learning capacity: sum of words correctly recalled during the five immediate learning trials (maximum score: 75 words). -Increased functional connectivity of the hippocampus in the frontal, parietal, and occipital areas. -Decrease in HB1ac (−0.13%) and DBP (−5.2 mmHg), and increase in leptin (+7.5 ng/mL). -No change in physical activity and mood.
		(0–6+1/2 months)	
Anton et al.(2018) [10]	-I: 1000 mg/day of resveratrol -I: 300 mg/day of resveratrol -C: placebo	-COWA. -Word Digits Forward and Backward and Digit Symbol Substitution Test (WAIS-IV tasks) -Eriksen flanking task- HVLT-R. -Change of tasks -TMT A and B	-Better psychomotor speed in the TMT A test in the intervention group with 1000 mg/day of resveratrol compared with 300 mg/day of resveratrol (−10.59 s) and placebo (−8.79 s). -No significant changes in performance on tests of visual attention, working memory, verbal fluency, and semantic memory between treatment groups.
		(0–3 months)	
Moran et al. (2018) [26]	-I: multi-ingredient supplement (omega-3 polyunsaturated fatty acids, vitamin D, 150 mg resveratrol and whey protein) -C: placebo	-TMT A and B -RAVLT -TUG -CFQ -Stroop test (CW version) -COWA -Digits of words forwards and backwards (WAIS-III task)	-In the intervention group there was a shorter response time in the Stroop Test both at 3 and 6 months, compared with the placebo group. -No significant changes in the rest of the tests.
		(0–3-6 months)	
Wong et al.(2016) [27]	-I: 300 mg/day of resveratrol -I: 150 mg/day of resveratrol -I: 75 mg/day of resveratrol -C: placebo	-CVR	-Resveratrol consumption increased CVR in MCA with all 3 doses of resveratrol: mean change from placebo was 13.8 ± 3.5% at intake of 75 mg resveratrol, 8.9 ± 3.5% in the 150 mg resveratrol dose, and 13.7 ± 3.3% in the 300 mg resveratrol dose. -The 75 mg dose was the only effective one in PCA, 13.2 ± 4.5% compared to placebo.
		(0–45/60 min–90/120 min after treatment). Four separate visits of 7 days were made	
Evans et al. (2017) [11]	-I: 150 mg/day of trans-resveratrol (2 capsules of 75 mg) C: placebo	-TMT A and B -RAVLT -Cambridge Semantic Memory Battery -BP and heart rate -TCD -CVR -Double Scope Task -POMS-V2 -CES-D	-Significant improvements in the immediate RAVLT (+4.5 words), in categorical fluency (+3.2 words), and in the camel and cactus test (+2.8 correct answers). These improvements correlate with a 17% CVR increase. -Mood tended to improve in the intervention group but not significantly.
		(0–3 + 1/2 months)	
Kobe et al. (2017) [13]	-I: 200 mg of resveratrol and 350 mg of quercetin daily -C: Placebo 1015 mg/day of olive oil	-RAVLT -PANAS -BDI -STAI X1 -Freiburger physical activity questionnaire -Analytical parameters (glucose, insulin, LDL, HDL, HbA1c and hs-CRP) -Anthropometric parameters (BMI and weight) -fMRI	-No significant differences in memory performance between the intervention group and the placebo group. -HbA1c was significantly reduced (−0.15%) after resveratrol intervention. -Hippocampal volume is preserved, and functional connectivity at rest of the hippocampus improves in the intervention group.
		0–6 + 1/2 months	

ADAS-COG: Cognitive subscale of the Alzheimer’s Disease Rating Scale; BDI: Beck’s Depression Inventory; BDNF: Brain Derived Neurotrophic Factor; BP: Blood Pressure; CBF: Cerebral Blood Flow; CBV: Cerebral Blood Volume; CDR: Cognitive Drug Research; CES-D: The Center for Epidemiological Studies Depression Scale; CFQ: Cognitive Failures Questionnaire; CMIT: Carotid Intima Media Thickness; COWA: Controlled Oral Word Association Test; CVLT: California Verbal Learning Test-II; CVR: Cerebrovascular Responsiveness; DBP: Diastolic Blood Pressure; fMRI: magnetic resonance imaging; FPG: Fasting Plasma Glucose; GDS: Geriatric Depression Scale; HbA1c: Hemoglobin A1c; HDL: High Density lipoprotein cholesterol; HDRS: Hamilton Depression Rating Scale; HF: High Flavanol; HOMA-IR índice: [insulina sérica en ayunas (mU/L) x glucosa plasmática en ayunas (mmol/L)]/22.5; hs-CRP: High Sensitive C Reactive Protein; HVLT-R: Hopkins Verbal Learning Test-Revised; IF: Intermediate Flavanol; IGF-1: Insulin Like Growth Factor 1; 8-Iso-PGF: 8-iso-prostaglandin F (índice de peroxidación lipídica relacionado con el estrés oxidativo); LDL: Low Density Lipoprotein Cholesterol; LF: Fow Flavanol; MCA: Middle Cerebral Arteries; MDA-LDL: Malondialdehyde-modified Low Density Lipoprotein;MFQ: Memory Function Questionnaire; MMSE: Mini Mental State Examination; MRS: Menopausal Rating Scale; NPI-Q: The Neuropsychiatric Inventory Questionnaire; PANAS: Positive and Negative Affect Schedule; PCA: Posterior Cerebral Arteries; POMS-V2: Profile of Mood States; RAVLT: Rey Auditory Verbal Learning Test; ROFC Test: Rey Ostereith Complex Figure Test; SPB: Systolic Blood Pressure; STAI X1: State Trait Anxiety Inventory; TC: Total Cholesterol; TCD: Transcranial Doppler; TG: triglycerides; TMT A y B: Trail Making Test A y B; TNF-: Tumor Necrosis Factor; TUG: Timed Up and Go; VFT: Verbal Fluency Test; WAIS-IV: Wechsler Adult Intelligence Scale IV; WAIS-III: Wechsler Adult Intelligence Scale III; WCST: Wisconsin Card Sorting Test.

**Table 4 antioxidants-11-01549-t004:** Summary assessments of risk of bias of the studies included with the Cochrane Collaboration’s tool.

	Generation of Random Sequences (Selection Bias)	Allocation Concealment (Selection Bias)	Participant and Staff Blinding (Performance Bias)	Blinding of the Outcome Assessment (Detection Bias)	Incomplete Outcome Data (Attrition Bias)	Selective Reporting (Reporting Bias)
Ide et al. (2016)	+	?	?	?	+	+
Mastroiacovo et al. (2015)	?	?	+	?	+	+
Brickman et al. (2014)	?	?	−	?	+	+
Desideri et al. (2012)	?	+	+	+	+	+
Krikorian et al. (2012)	?	?	+	?	?	?
Lamport et al. (2015)	+	+	+	+	?	?
Witte et al. (2014)	?	?	+	?	+	?
Pase et al. (2013)	+	+	+	+	?	?
Anton et al. (2018)	+	?	?	?	?	?
Moran et al. (2018)	+	?	+	+	+	+
Lee et al. (2017)	?	?	+	+	+	?
Wong et al. (2016)	?	+	+	+	+	+
Evans et al. (2017)	−	+	+	+	+	+
Kobe et al. (2017)	+	+	?	?	+	?

(+) Low risk of bias (?) Risk of unclear bias (−) High risk of bias.

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
