# Peer review of "Effect of Polyphenolic Complements on Cognitive Function in the Elderly: A Systematic Review"

_antioxidants, 2022, doi:10.3390/antiox11081549_

Round 1

Reviewer 1 Report

This is a systematic and comprehensive review on the effects of polyphenolic complements on cognitive function.   Basically, this manuscript is in well-written, excellent organization and structure, solid scientific background, clear on its research strategy and criteria.  The conclusive and discussion parts are in depth and send out explicit message to readers.  

The reviewer has no issue on this manuscript. 

Author Response

Thanks for reviewing the manuscript.

Reviewer 2 Report

Potential health benefits connected with the polyphenol-rich diet are often discussed in various contexts. Especially, bioavailability of the polyphenolics, their beneficial effects in cardiovasculatory system, chemopreventive activity and interactions with gut microbiota arouse interest. An influence of dietary polyphenols on cognitive the function in humans is less explored although very interesting field of research, taking into consideration the progressive aging of the Western population and a growing number of people struggling with dementia disorders. Despite the presence of the subject in the popular discourse, scientific data to support efficacy of polyphenol-rich diet in the prevention or retardation of the progress of dementia are sparse.

The review is a successful attempt to summarize the results of randomized controlled trials on human subjects, conducted with use of either isolated compounds or polyphenol-rich preparations and complex dietary supplements containing polyphenols. In my opinion the manuscript is valuable and worth publishing, however, some imperfections should be removed or corrected before the final acceptance.

Detailed comments:

Page 8, Table 3. “Mod Bent” test and “Moodbent” should be corrected into ModBent (modified Benton test).

Page 13, lines 154-156. “However, no articles related to cognitive impairment for compounds of the flavonol family were found.”

This is not true. Studies by Witte et al. [24] and by Kobe et al. [13] were conducted using 200 mg of resveratrol (stilbene) and 320 or 350 mg of quercetin (flavonol) daily. The studies were described as those investigating resveratrol effects. In fact, quercetin was the main active compound dosed.

Page 13, line 160. “incluyed” -> included

Page 13, lines 165-166. “For studies using resveratrol...” The studies used resveratrol and quercetin in mixture with resverartol.

Page 16, line 283. “Of the six studies with stilbenes, all of them using resveratrol...”; please correct the phrase, e.g.: “Of the six studies with stilbenes, four of them using resveratrol and two using flavonol (quercetin) with an admixture of resveratrol”.

Page 17, lines 352-353 and 357. “...supplementation with flavanols and flavonoids...” Flavanols are a subclass of flavonoids.

Conclusions have been duplicated in the terminal part of the Discussion section (lines 449-465). This part of the text should be removed.

Author Response

Thanks to the reviewer for helping to improve the manuscript.

Detailed comments:

Page 8, Table 3. “Mod Bent” test and “Moodbent” should be corrected into ModBent (modified Benton test).

Mod Bent test and Moodbent has been modified by ModBent in table 3.

Page 13, lines 154-156. “However, no articles related to cognitive impairment for compounds of the flavonol family were found.”

This is not true. Studies by Witte et al. [24] and by Kobe et al. [13] were conducted using 200 mg of resveratrol (stilbene) and 320 or 350 mg of quercetin (flavonol) daily. The studies were described as those investigating resveratrol effects. In fact, quercetin was the main active compound dosed.

The sentence has been changed to the following: However, no randomized clinical trial performed exclusively with compounds belonging to the flavonol family and their influence on cognitive impairment was found. Two of the works found were conducted using together compounds from the family of flavonols and stilbenes; 200 mg of resveratrol (stilbene) and 320 or 350 mg of quercetin (flavonol) daily [24, 13].

Page 13, line 160. “incluyed” -> included

The word has been corrected

Page 13, lines 165-166. “For studies using resveratrol...” The studies used resveratrol and quercetin in mixture with resveratrol.

And studies used resveratrol and quercetin in mixture with resverartol

The sentence has been changed in the manuscript.

Page 16, line 283. “Of the six studies with stilbenes, all of them using resveratrol...”; please correct the phrase, e.g.: “Of the six studies with stilbenes, four of them using resveratrol and two using flavonol (quercetin) with an admixture of resveratrol”.

The sentence has been modified.

Page 17, lines 352-353 and 357. “...supplementation with flavanols and flavonoids...” Flavanols are a subclass of flavonoids.

This paragraph has been modified.

Conclusions have been duplicated in the terminal part of the Discussion section (lines 449-465). This part of the text should be removed.

Duplicate phrases have been removed in the part of the Discussion section.